# Dissecting the circuit for blindsight to reveal the critical role of pulvinar and superior colliculus

Masaharu Kinoshita [1,2], Rikako Kato[1,3], Kaoru Isa[1,3], Kenta Kobayashi[4,5], Kazuto Kobayashi[6], Hirotaka Onoe[7,8] & Tadashi Isa[1,3,4,5,7]

In patients with damage to the primary visual cortex (V1), residual vision can guide goal-directed movements to targets in the blind field without awareness. This phenomenon has been termed blindsight, and its neural mechanisms are controversial. There should be visual pathways to the higher visual cortices bypassing V1, however some literature propose that the signal is mediated by the superior colliculus (SC) and pulvinar, while others claim the dorsal lateral geniculate nucleus (dLGN) transmits the signal. Here, we directly test the role of SC to ventrolateral pulvinar (vlPul) pathway in blindsight monkeys. Pharmacological inactivation of vlPul impairs visually guided saccades (VGS) in the blind field. Selective and reversible blockade of the SC-vlPul pathway by combining two viral vectors also impairs VGS. With these results we claim the SC-vlPul pathway contributes to blindsight. The discrepancy would be due to the extent of retrograde degeneration of dLGN and task used for assessment.

[1] Department of Developmental Physiology, National Institute for Physiological Sciences, Okazaki 444-8585, Japan. [2] Department of Physiology, Hirosaki University School of Medicine, Hirosaki 036-8562, Japan. [3] Department of Neuroscience, Graduate School of Medicine, Kyoto University, Kyoto 606-8501, Japan. [4] Department of Viral Vector Development, National Institute for Physiological Sciences, Okazaki 444-8585, Japan. [5] School of Life Sciences, the Graduate University of Advanced Studies (SOKENDAI), Hayama 240-0193, Japan. [6] Department of Molecular Genetics, Institute of Biomedical Sciences, Fukushima Medical University, Fukushima 960-1295, Japan. [7] Human Brain Research Center, Graduate School of Medicine, Kyoto University, Kyoto 606-8501, Japan. [8] Division of Bio-Function Dynamics Imaging, RIKEN Center for Life Science Technologies (CLST), Kobe 650-0047, Japan. Correspondence and requests for materials should be addressed to T.I. (email: isa.tadashi.7u@kyoto-u.ac.jp)

Patients with damage to the primary visual cortex (V1) show loss of visual awareness in the affected visual field, but some of them can still exhibit the capacity of residual vision by which they can direct the hand or eyes to the visual targets in the blind field[1–3]. Such dissociation of ability for the visually guided control of goal-directed movements and visual awareness has been termed blindsight and attracted the attention of many neuroscientists, psychologists, philosophers, clinicians, etc. However, the neural mechanism of blindsight has been still controversial. In an earlier study, it was attributed to the visual pathway mediated by the superior colliculus (SC), the pulvinar (Pul), and the extrastriate visual areas[4]. The role of SC in blindsight has been confirmed based on the inactivation or lesion studies in monkeys with V1 lesion[5–7] and by manipulation of visual stimulus parameters in human blindsight patients[8–11]. However, as to the role of Pul, Kaas and colleagues suggested that it should be minimal, because the tecto-recipient zone of the Pul minimally overlaps the areas with the area containing the neurons projecting to the extrastriate cortex[12]. On the other hand, Cowey and Stoerig[13] showed that there are surviving neurons in the dorsal lateral geniculate nucleus (dLGN), which project to the extrastriate cortex, after the V1 lesion and suggested that they would mediate the visual inputs directly to the extrastriate cortex in blindsight monkeys. It was later shown that a part of konio-cellular layer neurons (K-cells) in the dLGN[14] directly project to the area middle temporal area (MT)[15]. In line with this, the study by Schmid et al[16]. showed that inactivation of the dLGN impaired visual responses in the extrastriate visual areas and performance of visually guided saccades (VGSs) to the targets on the scotoma in the monkeys with V1 lesion. Yu et al.[17] showed that in the marmosets with partial V1 lesion, the remaining dLGN neurons might have a potential to convey information for the residual vision. More recent patient studies showed further evidence suggesting that human blindsight is mediated by the LGN–MT pathway[18,19]. Thus multiple evidence has been accumulated proposing that the LGN–MT pathway plays a major role in blindsight. However, a study using the trans-synaptic retrograde tracing technique showed that there exists a pathway from SC to MT or parietal cortex via the Pul[20,21]. Berman and Wurtz[22] showed that some Pul neurons mediate inputs from the SC to the area MT in V1-intact monkeys. Furthermore, Bender[23,24] suggested that V1-recipient Pul neurons might have become tecto-recipient neurons after >3 weeks from the V1 lesion. On the other hand, the role of retina–Pul–MT pathway in the visual processing has been demonstrated in the common marmosets with early-life primary visual cortex lesions[25]. However, the contribution of the Pul and the SC–Pul pathway to blindsight in adult subjects has never been directly tested. Therefore, in this study, to reveal a role of the SC–Pul pathway in blindsight, we first inactivated the ventrolateral Pul (vlPul) and tested the performance of VGS in unilateral V1-lesioned monkeys. And, as the next step, we selectively blocked the SC–vlPul pathway by using the double viral vector injection method, which was developed in our laboratory[26,27]. The results show not only the essential role of vlPul but also the critical contribution of the SC to vlPul pathway in the control of VSGs in blindsight monkeys. The results will be discussed in line with a recent proposition that blindsight is not a unitary phenomenon but is a constellation of functions mediated by multiple pathways remaining after the V1 lesion[28].

## Results

Two Japanese macaque monkeys (C and H) were trained to sit on a monkey chair and perform a VGS task. After completion of training, the unilateral V1 (right side in both monkeys) was aspirated under anesthesia with isoflurane. As shown in Fig. 1a–c,

the lesion included virtually the whole V1 (not only the occipital surface but also the calcarine fissure) and was ablated in Monkey-C, while in Monkey-H, the area representing the lower visual field partly remained (around 240°, Fig. 1c). Performance of the VGS task with high contrast stimuli recovered to about 90% level within approximately 2 months[29]. However, the decrease of contrast sensitivity was observed in the contralesional visual field (Fig. 1b–e), especially to the area representing the damaged site, as reported in our previous report[30]. At the lower contrast, both monkeys showed <40% correct saccades to the contralesional visual field (Fig. 1d, e). The present experiments were initiated 33 and 98 months after the V1 lesion in the Monkey-C and -H, respectively.

**Pul inactivation.** First, to examine whether the Pul is essential for VGS in blindsight monkeys, we performed a reversible blockade of the Pul using microinjection of muscimol, a GABA_A receptor agonist. To determine the injection site, we first mapped the ipsilesional Pul with extracellular recordings of single- or multi-unit responses to stimulation of the SC on the same side (stimulus intensity <1.2 mA). As shown in Fig. 2 and Supplementary Figure 6, orthodromic unit responses were found in the lateral and ventral part of the Pul (vlPul) with 1.3–5.1 ms latencies ($n = 46$) with small jitter, presumably through mono-synaptic linkage as previously shown by Berman and Wurtz[22]. Based on this finding, the injection sites of muscimol were determined in the vlPul (Fig. 3c, Supplementary Figures 2A and 2G). We carefully decided the muscimol injection sites to avoid inactivation of SC or dLGN. After microinjection of muscimol (concentration 5 μg/μL) into in the vlPul (0.2–0.5 μL for each site, 6 sites in Monkey-C and 3 sites in Monkey-H), the location of the injection site was confirmed by Gd$^{3+}$-based magnetic resonance image (MRI) contrast agent, which was included in the muscimol solution as shown in Fig. 4 and Supplementary Figure 4. The center of strong MRI signal was observed in the vlPul but not in the dLGN or SC. After the muscimol injection, performance of VGS task to the contralesional visual field, affected by V1 lesion, was severely impaired. As shown in Fig. 3 and Supplementary Figure 2, the success ratio was significantly reduced, and the reaction time (defined as time between the target appearance and the start of the saccade, see Methods for detail) was prolonged for the successful saccades to the contralesional visual field in both monkeys, while the injection of saline did not alter the responses. In Monkey-H, muscimol injection had no effect at the specific location (Supplementary Figure 3), which might correspond to the preserved visual field in contralesional side (around 240° in Fig 1c, e). These results strongly suggested that the vlPul plays a critical role for the performance of VGS task to targets in the visual field affected by V1 lesion.

**Selective block of the SC–vlPul pathway.** In addition to the previous study that inactivated the SC, we showed inactivation of the vlPul also impaired residual vision after V1 lesion. However, it is not yet demonstrated that the direct pathway from the SC to vlPul contributes to the blindsight. As briefly shown in Fig. 5 (see Methods and Supplementary Figure 5 for detail), to demonstrate the role of the SC–vlPul pathway in the blindsight, we adopted the pathway-selective and reversible blocking technique using the two viral vectors[26,27]. We first injected the highly efficient retrograde gene transfer vector, HiRet-TRE-eGFP.eTeNT into the vlPul (0.25–0.8 μL at 32 sites in Monkey-C and 31 sites in Monkey-H, see Supplementary Figures 6D and 6H, respectively). These injections were made at least 2 mm apart from the SC. Six to 7 days later, the second vector, AAV1-CMV-rtTAV16 was injected into the SC (0.3–0.6 μL at 8 sites in Monkey-C and

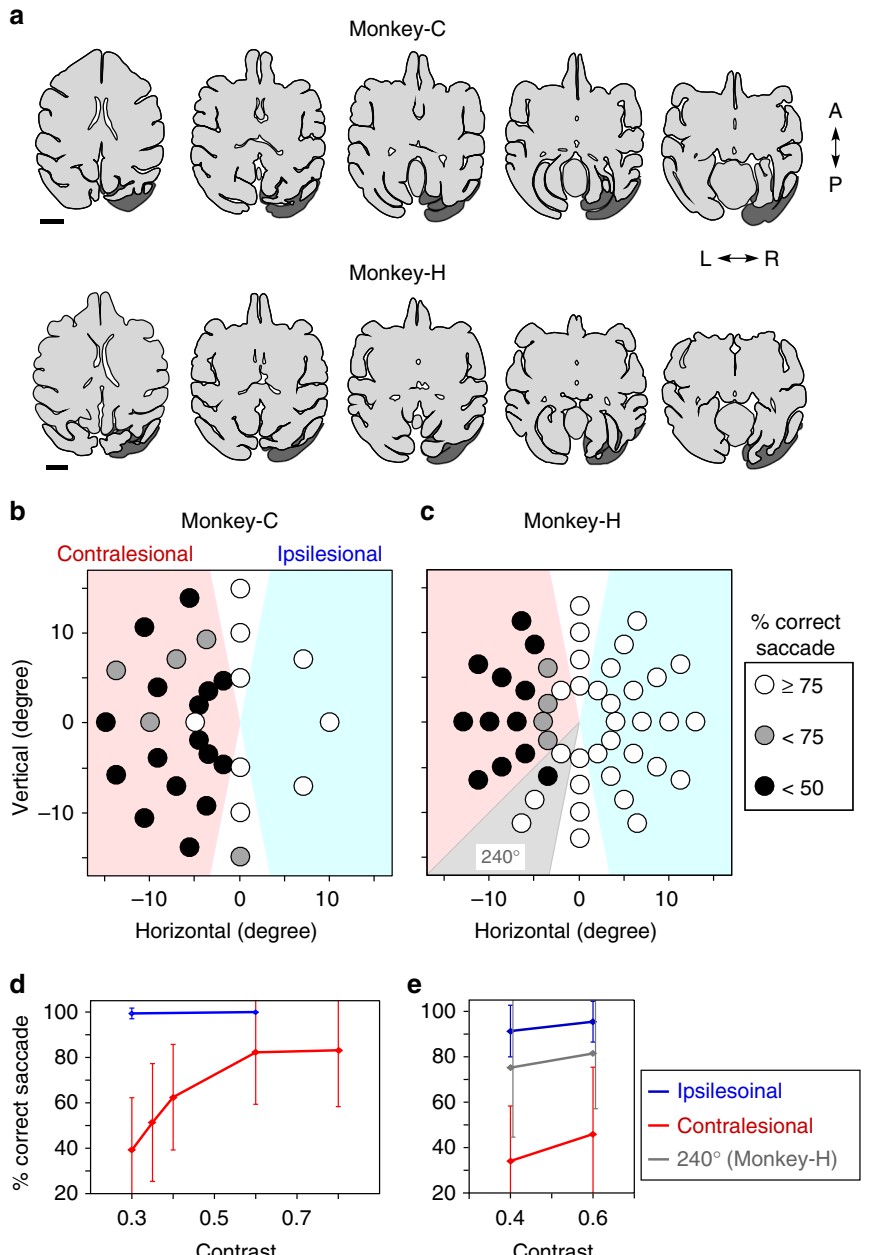

**Fig. 1** Unilateral V1 lesion and the affected visual fields. **a** Extent of V1 lesions: drawings of horizontal brain sections traced from the magnetic resonance images (MRI; shown in Supplementary Figure 1). The dark gray areas indicate the estimated lesion sites (see Methods). Note that lesion covered not only the occipital surface but also the calcarine fissure. Scale bar: 1 cm. **b**, **c** The deficit maps: percent correct saccades were plotted. The Michelson contrast of the target was 0.3 for Monkey-C and 0.4 for Monkey-H. The performance for 5 days was averaged (see Methods). **d**, **e** Contrast sensitivity: average of percentage of correct saccade to the targets in ipsilesional (blue) or contralesional visual field (red) was plotted against contrast intensity. For Monkey-H, data at 240° (gray) was excluded from the contralesional field. The performances for 5 days were averaged (see Methods). The error bars represent standard deviation (SD)

10 sites in Monkey-H). Six to 11 weeks after the second vector injection, oral administration of doxycycline (Dox) was initiated. During the Dox administration, the accuracy of saccade declined (Fig. 6b, h) than before administration (Fig. 6a, g). To quantify the effect of Dox administration on VGS performance, we analyzed the size of saccade error (defined as distance from saccade target to saccade endpoint) and the reaction time (see Methods and Fig. 6). In both monkeys, about 10 days after the initiation of Dox administration, the size of saccade error to particular targets in the contralesional visual field significantly increased (P < 0.01, t test, Fig. 6c, i). In addition, the reaction time of successful

saccades toward these targets was significantly prolonged (Fig. 6e, k). Such an increment of saccade error and prolonged reaction time were found widely in the contralesional visual field (Fig. 6d, f, j, l). As seen in these affected maps, the effect of double infection method did not cover all tested target locations; however, affected locations were largely overlapped between two indices and across different target contrast in each monkey (Supplementary Figure 7).

**Retrograde degeneration of dLGN.** The hypothesis against the contribution of Pul has been based on the assumption that the K-

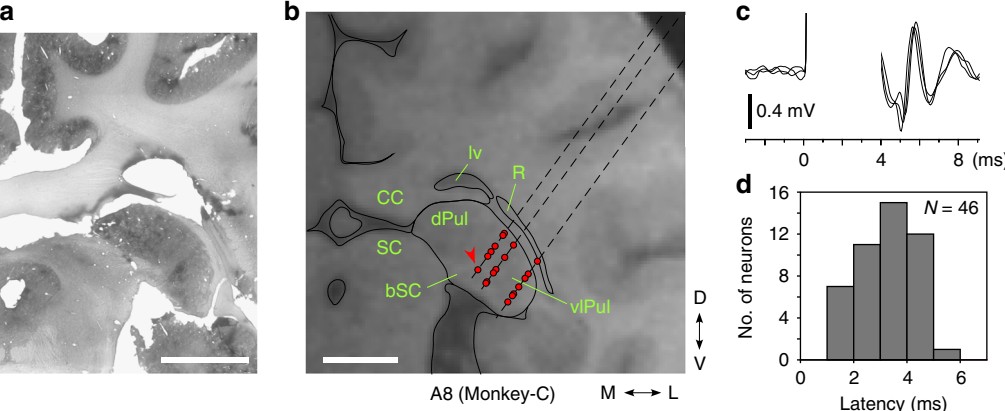

**Fig. 2** Location of the Pul neurons activated by the microstimulation of the SC. **a** Nissl-stained section corresponded to the MRI section shown in **b**. **b** The location of neurons (red dot), which showed presumably mono-synaptic responses to the microstimulation of the superficial layer of SC (see Methods and Supplementary Figure 6). Red arrowhead indicates the location of neuron shown in **c**. vlPul: ventrolateral pulvinar. dPul: dorsal pulvinar. bSC: brachium of superior colliculus. R: thalamic reticular nucleus. lv: lateral ventricle. CC: corpus callosum. A8: 8 mm anterior from the ear bar. Dashed line indicates electrode track (see Supplementary Figure 6). Scale bar: 5 mm. **c** Representative evoked spikes of the Pul neuron (arrowhead in **b**). Three consecutive traces are superimposed. The evoked spikes showed short latency and small jitter. **d** The distribution of response latencies of neurons, which showed presumably mono-synaptic response, recorded in the ipsilesional vlPul of two monkeys

cells in the dLGN directly transmit the visual signal to the area MT of the cortex. However, by Nissl staining we have observed that shrinkage of the dLGN and most of relatively large dLGN neurons were eliminated presumably owing to the retrograde degeneration of magno and parvo cells caused by the V1 lesion (Fig. 7a–d). Next, we tried to investigate the extent of degeneration of K-cells in the dLGN by anti-calmodulin-dependent protein kinase IIα (CaMKII) immunostaining, which is known to stain this subtype of dLGN neurons[14] in Monkeys -C, -H, and an additional monkey (-A) by comparing the CaMKII-positive neurons in the dLGN on the ipsilesional and contralesional sides (Fig. 7e–g). As shown in Fig. 7g and Supplementary Table 1, the CaMKII-positive cells were reduced to 18–34% (34% in Monkey-C, 18% in Monkey-H, and 22% in Monkey-A, which was used for other experiments and was sacrificed 71 months after the V1 lesion) on the ipsilesional dLGN. On average, the monkey with the longer survival time after the V1 lesion tended to have a smaller number of remaining CaMKII-positive cells (Supplementary Table 1). There still remained a considerable number of the K-cells in the dLGN. However, a large population of the K-cells was retrogradely degenerated in the dLGN. The remaining CaMKII-positive cells looked swollen on the ipsilesional side (arrow in Fig. 7f, Monkey-C). Such cells that appeared to have swollen were also observed in both Monkey-H and -A.

## Discussion

The above results showed that vlPul and the direct pathway from the SC to vlPul is, at least partly, involved in mediating the residual vision for execution of VGSs in the V1-lesioned monkeys. The electrophysiological recordings showed that a considerable number of neurons in the vlPul are responsive to the inputs from the SC, and inactivation of the area containing these neurons resulted in impairment of VGS in the blindsight monkeys. We visualized the injection site of muscimol in the vlPul using the imaging of $Gd^{3+}$ by MRI (Fig. 4); however, the diffusion constants of muscimol and $Gd^{3+}$ are not necessarily the same, and therefore the possibility that muscimol diffused into dLGN still remains. Thus the contribution of the dLGN to the execution of VGS in the V1-lesioned monkeys could not be

completely excluded by the muscimol injection experiment alone. The target area of these tecto-recipient neurons in Pul is not clear in the present study, but previous literature on intact monkeys suggests that they are projecting to the extrastriate cortices, such as the area MT[21,22] or parietal cortex[20]. Furthermore, to demonstrate the role of the SC–Pul pathway, we applied the double viral vector technique, which enables the pathway-selective synaptic transmission blockade that we previously developed[26,27] to selectively shut down the transmission from the SC to vlPul. The effect was moderate partly because of its technical limitation, but we could clearly observe the effects on VGS toward some particular points in the contralesional visual field. In our previous studies on the spinal cord neurons[26,27], the effect of Dox administration emerged on the 1–3 days after the start. However, in the present study on the superficial layer neurons of the SC in the monkeys, it took about 10 days for the effects to appear after Dox administration. Thus the time needed for the emergence of effects of tetanus neurotoxin might depend on cell types, which we must take into account for future studies. As for evaluation of the expression of enhanced tetanus neurotoxin light chain (eTeNT), it was found that in cell cultures the number of the eGFP-eTeNT fusion molecules necessary for suppressing the transmission is much smaller than that for making fluorescence visible (Ken-ichi Inoue and Masahiko Takada, personal communication). Therefore, there should be many false-negative cells whose synaptic transmission was suppressed but which did not show enhanced green fluorescent protein (eGFP) expression sufficient to be visualized. Because this means eGFP is not a sufficient indicator of eTeNT expression, no histological data for eTeNT expression of Monkey-C or -H was shown in this article.

The SC is considered to play a critical role in the control of saccadic eye movements[31,32]. Then, is the present result only reflecting the impairment of the saccadic motor system? The previous studies have shown that inactivation of SC in intact monkeys delays latency of saccades but does not impair execution of saccades itself[33–35]. Other studies have shown that Pul neurons that receive the input from SC showed similar properties with visual neurons in SC and exhibited no or weak pre-saccadic activity[36,37]. Bender and Baizer[38] have shown that saccade initiation, latency, and amplitude were not impaired in the

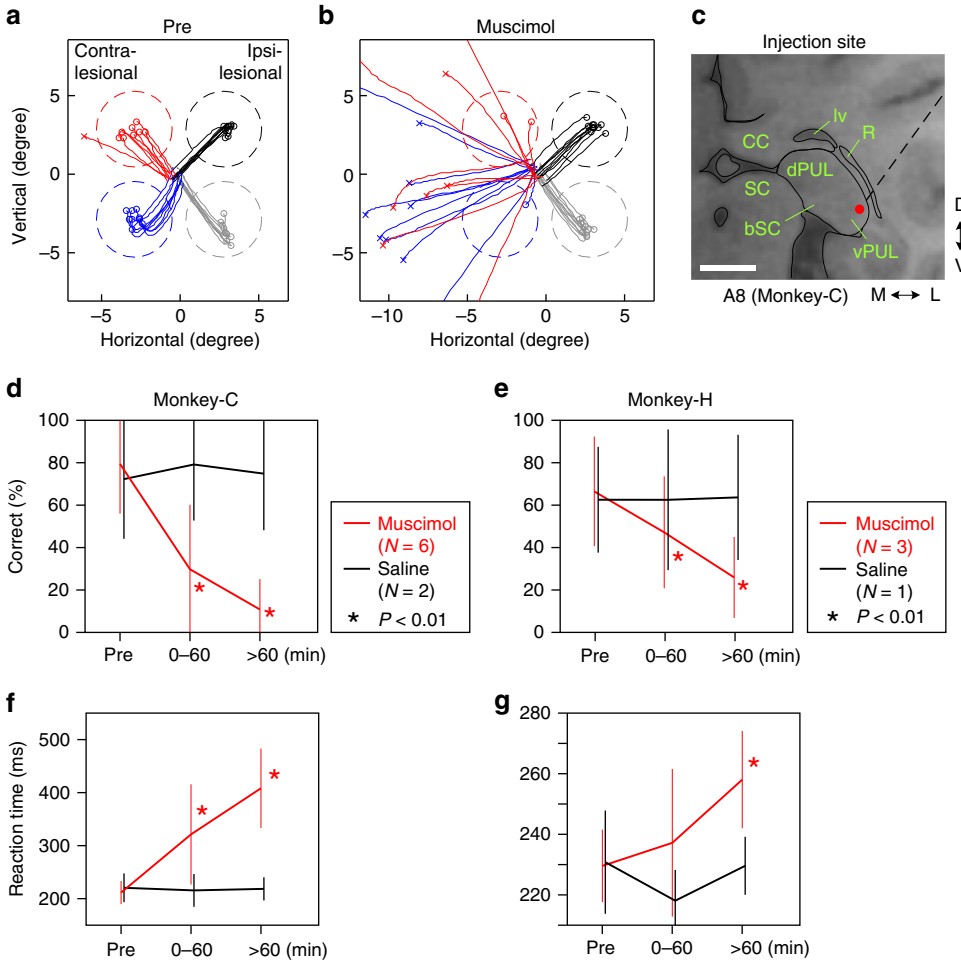

**Fig. 3** Inactivation of the vlPul impaired the VGS. **a**, **b** Representative trajectories of saccades in the VGS task before (**a**) and 60 min after (**b**) the injection of 0.5 μL muscimol. The target was presented at the center of the dashed circle, which indicates the target window. The saccade endpoint was depicted as o (correct saccade, inside of the window) or x (fail, outside). Different color corresponds to the difference of target location. The target contrast was 0.85. **c** Red dot indicates the injection site for **b**. The dashed line indicates the penetration of the injection needle. Scale bar: 5 mm. **d**, **e** The averaged ratios of the correct saccade for each time window (Pre: before injection. 0–60: 0 to 60 min after the injection. >60: after 60 min. Red line: muscimol injection. Black line: saline injection). The correct ratios were significantly decreased in muscimol injections (asterisk: $P < 0.01$, $t$ test, compared with pre) in both monkeys. No significant difference was found in saline injections ($P > 0.05$, $t$ test). The number of days used for the average is indicated as $N$ in each inset. Error bar indicates SD. The target contrast was 0.85 for Monkey-C and 0.6 for Monkey-H. Number of trials for each target in each time window were 4–11 for Monkey-C and 7–23 for Monkey-H. Number of targets for the average were 21 for Monkey-C and 12 for Monkey-H. Non-averaged results are shown in Supplementary Figure 2. **f**, **g** The averaged VGS reaction time is plotted in the same manner as **d**, **e**, respectively. The prolonged reaction time was observed after the muscimol injection in both monkeys

monkeys with Pul lesion. These observations suggest that the present results, which showed impairment of VGS by the inactivation of tecto-recipient zone of the Pul, could not be interpreted as the results of impairment of saccade motor system, but rather, Pul conveys critical visual information to the saccade motor systems in blindsight monkeys.

As described above, contribution of Pul and SC–Pul pathway was demonstrated. Then, what is the reason for the difference from the preceding studies by Schmid et al.[16], who claimed that the dLGN plays a major role in blindsight? One of our interpretation is that, in the Schmid et al. study, the lesion of V1 was limited to the gray matter of a part of the unilateral V1. In this case, therefore, the major part of the dLGN was supposed to be still remaining. In contrast, in our animals, the V1 lesion was made with extensive aspiration and included the white matter and partly the adjacent V2 (see Methods), and the behavioral test was performed as long as 33–98 months after the lesion. The Nissl staining of the dLGN in Monkey-C and -H revealed that a large

portion of the dLGN had degenerated on the ipsilesional side (Fig. 7a–d). The dLGN of the Monkey-C and -H was further processed on the immunohistochemical staining with anti-CaMKII antibody (Fig. 7e, f), which can visualize the K-cells in the dLGN projecting to the area MT[14,15]. As shown in Fig. 7g and Supplementary Table 1, 34% and 18% of the K-cells had survived compared to the contralesional side in Monkey-C and -H, respectively. Although not used in this behavioral experiment, the anti-CaMKII immunohistochemistry in another unilateral V1 lesion monkey (Monkey-A, 71 months after the lesion) showed that 22% of the K-cells had survived. It has been shown that V1- and MT-projecting dLGN neurons are segregated[15], although it is not known how many of the K-cells project to the V1. It is not clear whether the extensive lesion in our monkeys included the white matter and involved the axons from dLGN to MT. However, we would like to emphasize that the previously reported blindsight patients (such as the patient G.Y., e.g., Barbur et al.[39]) exhibited extensive damage to the V1 including the white matter,

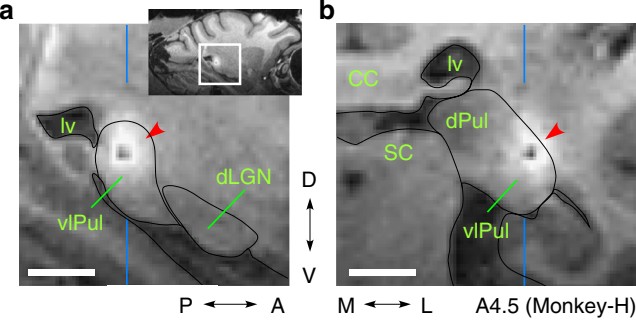

**Fig. 4** The diffusion extent of injected muscimol. Injection of mixture of muscimol and Gd$^{3+}$ (see Methods and i in Supplementary Figure 2G) resulted in the localized signal in the MRI (red arrowhead). It seems as if there was a black hole, but it was not a hole (Supplementary Figure 4). **a** The parasagittal slice: vertical blue lines indicate the anteroposterior position of **b**. Note that the strong signal did not extend to the dLGN. **b** The coronal slice: vertical blue lines indicate the mediolateral position of **a**. Scale bar: 5 mm

which suggests that our animal model is closer to the actual blindsight patients than the studies with partial V1 lesions. It is likely that the remaining K-cells in this study were projecting to the MT, and interestingly, the morphological changes were observed in the K-cells, which survived the lesion (Fig. 7f). This suggests that plastic changes occurred in the surviving K-cells to compensate for the degenerated neurons. Such plastic changes might lead to strengthening of existing subcortical–cortical pathways bypassing V1 or forming entirely new ones and larger scaled network on the affected hemisphere[40,41]. In our animals, the possibility of the damage to the dLGN–MT pathway could not be excluded; however, the present results clarified that, even under such a situation, Pul played a significant role in controlling VGS in the blindsight monkeys.

Altogether, it seems likely that both dLGN and Pul could be involved in blindsight. This is in line with the suggestion made by a recent review by Tamietto and Morrone[28] that blindsight would not be a unitary phenomenon but should be considered as a constellation of functions persisting after the lesion of V1, that is, multiple pathways can sustain different blindsight functions, which was behaviorally outlined by Danckert and Rossetti[42]. In this regard, it should be noted that Schmid et al.[16] and Ajina et al.[18,19] mainly used the moving visual stimuli to get their functional MRI data and proposed the dLGN–MT contribution, while our present study used VGS task and proposed the SC–Pul contribution. On the other hand, the researchers working on the process of biologically salient stimulus such as face and snakes suggest the involvement of SC–Pul–amygdala pathway in both humans and macaques[11,43–46]. These arguments are further supported by a recent study of a quantitative meta-analysis of neuroimaging data available for human blindsight[47]. Therefore, for the future study, it would be necessary to conduct double dissociation experiments to compare the effects of inactivation of dLGN or Pul in the same animal. Furthermore, in the present experiments, we used the VGS task both for the assessment of the visual perception and visuo-motor transformation process. For the future study, if possible, different task sets including the manual response task and a variety of visual or visuo-motor tasks would be necessary.

A number of experimental studies using the V1-lesioned macaque monkeys[5–7,16,29,30,48] have been performed. It was demonstrated that visual awareness is impaired in these animals as in the human blindsight patients: first by testing the behavior in Yes–No detection task[49], and more recently by applying the

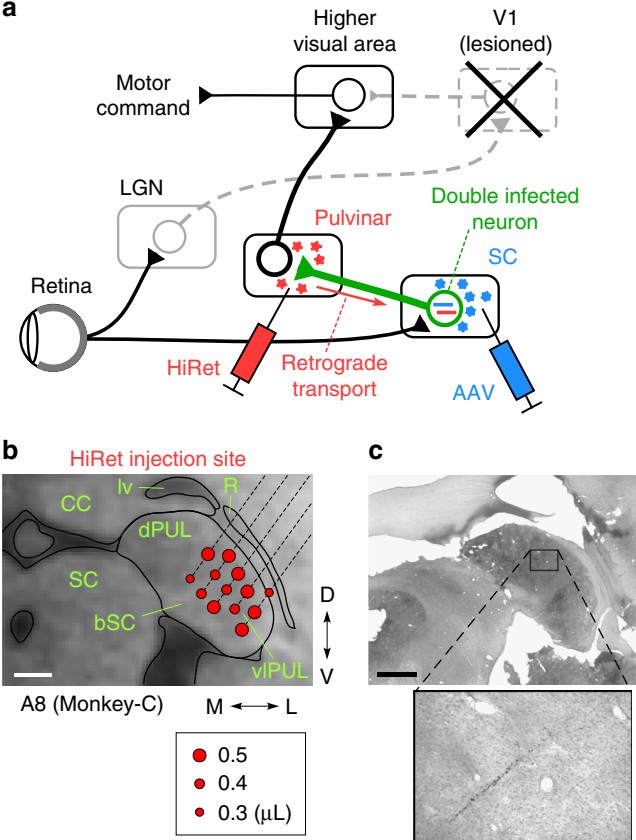

**Fig. 5** Selective blockade of the SC–vlPul pathway by using two viral vectors. **a** The retrograde gene transfer viral vector (HiRet-TRE-eGFP. eTeNT) was injected into the vlPul and the anterograde viral vector (AAV1-CMV-rtTAV16) was injected into the SC. Then double infection occurs only on the neurons whose cell bodies were in the SC and whose axons projected to the vlPul. The synaptic transmission from SC neuron to vlPul is selectively blocked under the administration of Dox. See Supplementary Figure 5 and Methods for detail. **b** Representative locations of the HiRet injections. Red dot indicates the injection site and volume (inset). See Supplementary Figure 6 for all the injections. **c** Above: Nissl-stained histological section corresponding to **b**. Below: magnified area showing the most dorsal injection track. Black rectangle indicates the magnified area. Scale bar: 2 mm

signal detection theory to examine the reduction in detection sensitivity in these animals[50]. In addition, recent studies have revealed that blindsight is not a simple non-conscious reflexive action, but as long as simple visual cues are used, significant level of action repertories that require more complex cognitive processing, such as short-term spatial memory[51] and associative learning[52], is sustained in the blindsight monkeys. These processes may involve forebrain areas (including higher association of cortices), thalamus, and basal ganglia. Clarifying the central visual pathway responsible for the residual vision and visual cognitive functions in blindsight would be expected to help us understand the neural substrate for the non-conscious cognitive processes with blindsight and presumably in the intact brain.

## Methods
**Animals**. Two male macaque monkeys (*Macaca fuscata* and *Macaca mulatta*, body weight 6.8 and 9.0 kg) were used in this study. Monkey-H is the same animal used in the previous study[29]. Monkey-A, which was used only for the immunohistochemistry in this study, is the same animal used in another study[30].

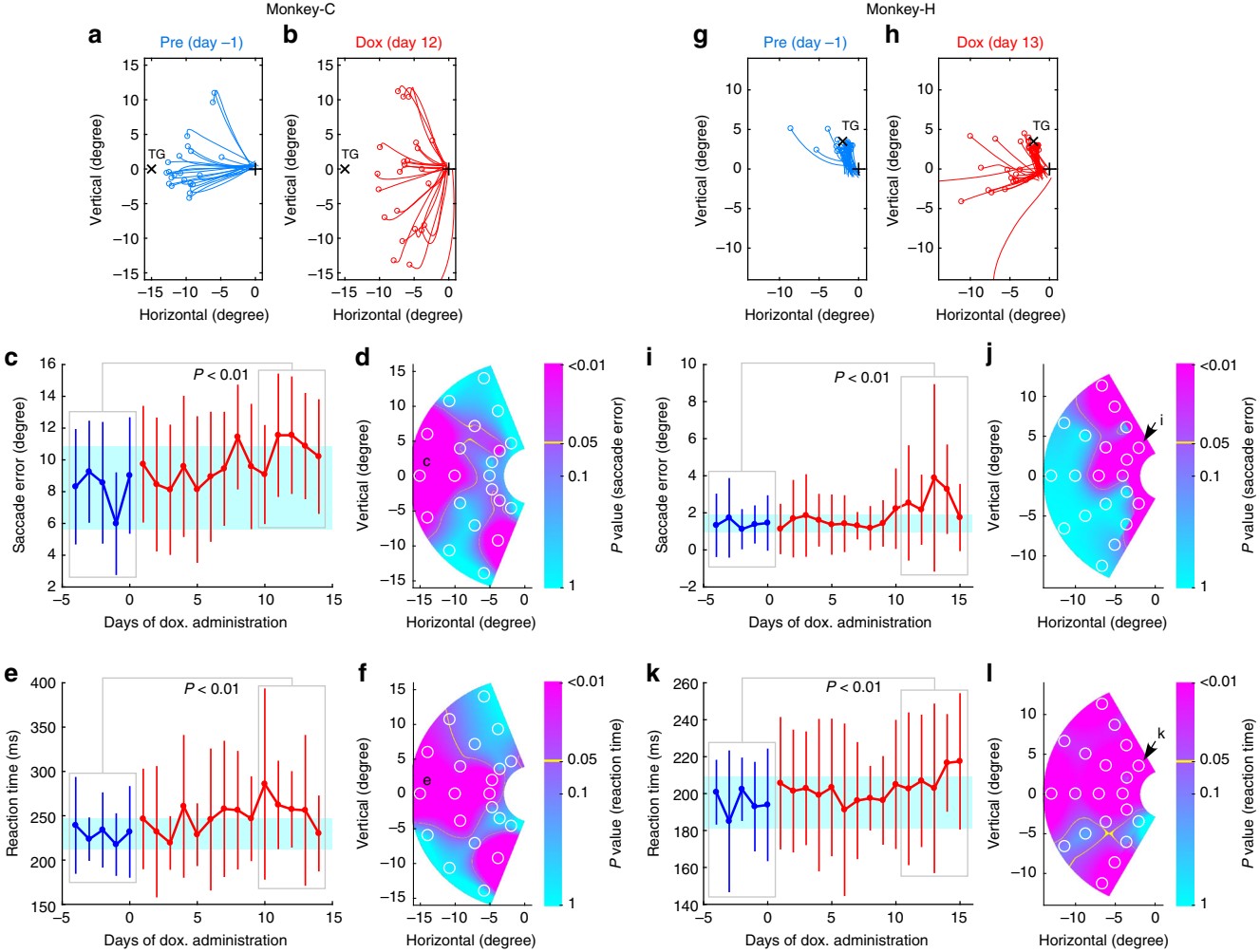

**Fig. 6** Selective blockade of the SC–vlPul pathway impaired the VGS. **a**, **b**, **g**, **h** Representative saccade trajectories and saccade endpoints before administration (Pre) and during Dox administration (Dox). The marks indicate the location of saccade endpoint (o), target (x), and fixation point (+). The size of saccade error (distance from target to saccade endpoint) increased in the Dox period in both monkeys. The target contrast was 0.4 for both monkeys (**a–f**, Monkey-C. G-L, Monkey-H). **c**, **i** For the representative target shown in **a** or **g**, the size of saccade error is plotted against Dox administration day. Blue: before administration. Red: during Dox administration. Dot: mean of saccade error of a day. Error bar: SD. Cyan: range between mean−2SD and mean+2SD of the saccade error in the Pre period. The number of trials in each day was 14–30 (**c**, **e**; Monkey-C) and 40–49 (**i**, **k**; Monkey-H). The saccade error increased around 10 days from start of Dox in both monkeys (P < 0.01, t test for set of data indicated by gray boxes). **d**, **j** P values of t test for the size of saccade error between Pre and late Dox (5-day data for each period, indicated by gray box in **c** or **i**) are mapped on the visual field with color code. For each target location, the number of trials included in 5-day data was 87–119 (**d**, **f**; Monkey-C) and 206–226 (**j**, **l**; Monkey-H). White circle indicates the target location for test. Representative target location shown in **c** or **i** is indicated by its letter on the map. The values are interpolated between tested locations. **e**, **k** For the same representative target, the reaction time (time between target appearance and start of saccade) of the VGS is plotted in the same manner as **c** and **i**. The reaction time increased in the late Dox period (P < 0.01, t test for set of data indicated by gray boxes). **f**, **l** P values of t test for the reaction time are mapped in the same manner as **d**, **j**

**Approval from the Institutes**. The animal experimental and surgical procedures in this study followed the guidelines of the National Institutes of Health and the Ministry of Education, Culture, Sports, Science and Technology (MEXT) of Japan, and were approved by the Institutional Animal Care and Use Committee of the National Institutes of Natural Sciences or by the Ethics Committee on Animal Care and Use of RIKEN Center for Life Science Technologies.

**Surgery**. All surgical procedures were performed under anesthesia, introduced with xylazine hydrochloride (2 mg/kg) and ketamine hydrochloride (5 mg/kg) and maintained with isoflurane (1.0–1.5%). An eye coil was implanted for monitoring the eye position in Monkey-H. A head holder was implanted for fixing the head position. Unilateral V1 (right side in Monkey-C and -H, left side in Monkey-A) was removed by aspiration. The extent of the lesion may include the extrastriate cortices (V2 or V3) because we attempted to remove V1 completely, except for the lateral part of V1 to preserve the foveal vision. To verify the extent of the lesion, MRIs of the brain were acquired after the V1 lesion (Fig. 1). The details of these procedures are described previously[7,29,30,51].

**Task**. Monkeys were trained for a VGS task before making V1 lesion. Monkeys were trained to sit in a primate chair with its head fixed. Eye movements were recorded by the magnetic search coil method[53] (MEL-25; Enzanshi Kogyo, Japan) for Monkey-H or by the video eye tracking system (Spot3; Hamamatsu Photonics, Japan) for Monkey-C. Visual stimuli were presented on the liquid crystal display (DELL 1704FPV, USA for Monkey-C; Mitsubishi RDT272WX, Japan for Monkey-H) positioned 31 cm (Monkey-C) or 42 cm (Monkey-H) from the monkeys' eyes. The visual stimuli and data acquisitions were controlled by the computer system (Tempo for Windows; Reflective computing, USA). Eye positions were sampled at 1 kHz for Monkey-H and at 250 Hz for Monkey-C. The VGS task began with the presentation of a fixation point. After the monkey kept fixation for 400–1200 ms on the fixation point, the saccade target was presented, and the fixation point disappeared simultaneously. When the monkey initiated a saccade at least 50 ms after the target appearance (to avoid an anticipatory saccade) and made a saccade into the target window before 115–130 ms from the start of that saccade (to avoid a double saccade) and kept a gaze for 200 ms, the monkey was rewarded with drops of juice. The reaction time was defined as the time between the target appearance and the start of the saccade. The start of the saccade was defined as the time point

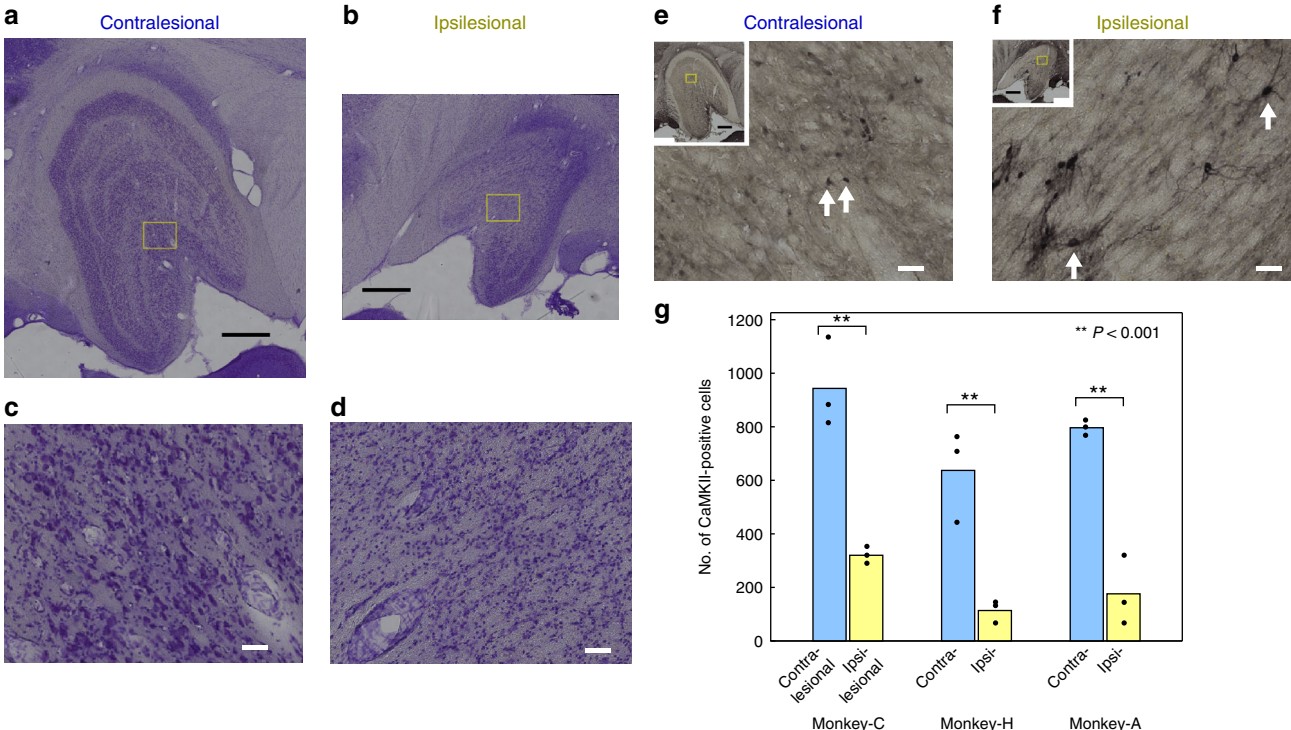

**Fig. 7** Degeneration of the dLGN on the side of extensive V1 lesion. **a–d** The representative Nissl-stained coronal section of dLGN of Monkey-C. **a**, **c** Contralesional. **b**, **d** Ipsilesional. **a–d** were taken from a single histological section. On the ipsilesional side, the size of dLGN was small (Supplementary Table 1), layer structure was unclear (**b**), and most of the large (10 μm order) cells were missing (**d**). **e**, **f** The representative anti-CaMKII immunostained dLGN (adjacent to the section of **a–d**). **e** Contralesional. **f** Ipsilesional. The CaMKII-positive cells look swollen on the ipsilesional side (arrow in **f**). Scale bars: 1 mm for **a**, **b** and insets in **e**, **f**, 50 μm for **c–f**. **g** Each bar represents the number of CaMKII-positive cells in the dLGN, averaged from three histological sections in each monkey (Coronal: Monkey-C and -H. Parasagittal: Monkey-A). Blue: on the contralesional side. Yellow: on the ipsilesional side. Non-averaged numbers are plotted (black dot) and shown in Supplementary Table 1. Monkey-A was not used in the main experiments of this article; however, V1 lesion of Monkey-A was made in a similar manner to Monkey-C and -H. Numbers of the CaMKII-positive cells on the ipsilesional side were only 34% (Monkey-C), 18% (Monkey-H), and 22% (Monkey-A) of the number of the contralesional side, respectively. In all monkeys, the numbers of cells were significantly different between the contralesional and ipsilesional sides (double asterisks: $P < 0.001$, $t$ test)

preceding the detected saccade at which the eye movement velocity exceeded 30°/s and the saccade endpoint was defined as the eye position at which the eye velocity declined <30°/s after the start of the saccade as previously reported[30]. In this article, we defined the size of saccade error as the distance from the saccade target to the saccade endpoint. The fixation point and the saccade target were small red spots with 0.4° in diameter and with luminance contrast 0.3–0.9 (Michelson contrast). The saccade target window sizes were varied depending on its eccentricity (for the eccentricity 4–15°, the window radius 2.4–8° for Monkey-C, 1.8–3.3° for Monkey-H). The saccade targets were consistently presented in the left (contralateral to lesioned V1, contralesional) visual field or in the right (ipsilesional) field in one experimental block. In each trial, location and contrast of the saccade target was pseudo-randomly chosen from a set of target conditions (8–32 locations and 2–4 contrasts in each block).

**Estimation of the extent of V1 lesion**. Before the experiments, the MRIs were taken after 33 months (Monkey-C) or 98 months (Monkey-H) from the V1 lesion. The extent of V1 lesion (Fig. 1a, dark gray area) was reconstructed by the mirror-reversed image of the intact hemisphere, which was transformed to fit in the locations of preserved sulcus in the lesioned hemisphere. Sensitivity plots in the affected visual field was obtained by measuring the success rate of VGS toward the target with varying luminance (Fig. 1b–e).

**Mapping the SC and Pul**. Before the muscimol and viral vector injections, we mapped the superficial layer of SC and the region of the Pul that received inputs from the superficial SC. First, we penetrated a tungsten microelectrode (FHC Inc, USA) into the SC by passing through a metal guide tube that was fixed in a 1-mm grid in the recording chamber (Crist Instrument Co Inc, USA) that was tilted 40° to posterior. We used the micromanipulator to advance the microelectrode (MO-97A; Narishige, Japan). We recorded neural responses in the SC while the monkey performed the VGS task and mapped the response field and the depth profile in the SC. Next, we placed the electrode at the center of the visually responding zone

(typically 0.7–1.2 mm depth from the dorsal surface of the SC). Then we penetrated another microelectrode into the Pul by passing through a guide tube with 1-mm grid fixed in another chamber that tilted 30° laterally. The electrical stimulation was applied through the electrode placed at the superficial SC (typically 0.1–1.2 mA, 100–120 μs pulse width, single biphasic pulse). We recorded extracellular responses in and around the Pul and mapped the sites that showed short latency responses (1.3–5.1 ms with small jitter) elicited by the SC stimulation. The short latency responding areas were mainly distributed in the vlPul (Fig. 2 and Supplementary Figure 6), which was consistent with the previous reports[21,22].

**Injection of muscimol**. We injected 0.2–0.5 μL muscimol (5 μg/μL) at 0.1 μL/min into the vlPul (Supplementary Figure 2) and recorded the behavioral performance to the VGS task before and after the muscimol injection. To determine the site of muscimol injection, we injected the mixture of muscimol and gadodiamide hydrate (0.04 μmol/μL, Omniscan; Daiichisankyo, Japan), which contains the magnetic resonance contrast agent $Gd^{3+}$, in some experiments. MRI was performed just after the recordings of the behavior test (90–130 min after the injection). In the control experiments, 0.5 μL saline was injected at the same site in the same manner as muscimol injection experiments. For the injection of muscimol into the vlPul, we used the metal needle (22028–01, pt2; Hamilton, Switzerland) that was connected to a perfluoroalkoxy tube and a micro syringe. The metal needle was attached to the same micromanipulator used for the electrophysiological experiments and advanced by passing through the guide tube and the grid in the same manner.

**Injection of viral vectors**. For the pathway selective blockade of the synaptic transmission from the superficial SC to vlPul, we applied the genetic dissection method using the double viral vectors on this pathway[26,27] (Fig. 5a). As shown in Supplementary Figure 5, 33 months (Monkey-C) or 98 months (Monkey-H) after the unilateral V1 lesion, the highly efficient retrograde gene transfer lentiviral vector (HiRet), which carried eTeNT under the tetracycline responsive element (TRE) (HiRet-TRE-eGFP.eTeNT, titer $4.7 \times 10^{11}$ copies/mL, 0.25–0.55 μL/site,

32 sites in 14 penetrations for Monkey-C; titer 0.9–1.2 × 10$^{12}$ copies/mL, 0.6–0.8 µL/site, 31 sites in 21 penetrations for Monkey-H) was injected into the vlPul, which received (presumably) mono-synaptic input from the SC (Fig. 2 and Supplementary Figure 6). Six to 7 days later, adeno-associated virus serotype 1 (AAV1), which carried enhanced reverse tetracycline transactivator (rtTAV16) (AAV1-CMV-rtTAV16; titer 8.0 × 10$^{11}$ viral genomes (v.g.)/µL, 0.3–0.6 µL/site, 8 sites in 8 penetrations for Monkey-C; titer 4.0 × 10$^{10}$ v.g./µL, 0.6 µL/site, 10 sites in 10 penetrations for Monkey-H) was injected into the SC. These injections were performed in the same way as the muscimol injections described above. Then double infection of these vectors would happen only on the neurons with cell bodies in the SC and axons projected to the Pul.

**Dox administration.** As shown in Supplementary Figure 5, 6–11 weeks after the vector injections, oral administration of Dox (15–25 mg/kg/day for Monkey-C; 25 mg/kg/day for Monkey-H) was started and continued for 14–23 days. For Monkey-C, 90 days after the offset of Dox administrations, the administrations were repeated for 25 days. In the double infected neurons, Dox would trigger the expression of eTeNT, which would suppress release of neurotransmitter. That is, the neural transmission of the double infected SC neurons projecting to the vlPul is selectively blocked. Before and during Dox administration period, the behavioral performances in the VGS task were recorded.

Before the second round of Dox administration for Monkey-C, the ratio of correct saccade was considerably deteriorated (Supplementary Figure 8) even though the target contrast was higher than the first round, presumably due to loss of motivation because of absence of training for 3 months after the first round. The experiments were restarted after the period without sufficient training and the data for the second round was obtained; however, retrospectively we decided to exclude the data from the second round from the analysis to avoid using such data as the control.

**Histology.** After the Dox administrations for 23–25 days, monkeys were deeply anesthetized with intravenous injection of sodium pentobarbital (50–100 mg/kg) and transcardially perfused with 0.05 M phosphate-buffered saline (PBS) and then 4% paraformaldehyde in 0.1 M phosphate buffer (pH 7.4). The brains were sectioned at a thickness 40 µm for coronal slices for Monkey-C and -H or 50 µm for parasagittal slices for Monkey-A. The sections of dLGN were immunostained with anti-calmodulin-dependent protein kinase IIα antibody (Affinity BioReagents, Golden, CO, USA) modified from the manufacturer's instructions. They were washed in PBS with 0.1% Triton-X followed by incubation in 50% ethanol. After washing the sections, they were dipped in the blocking solution (PBS with 2% normal goat serum, 0.1% Triton X-100, 0.5% fish gelatin, 0.5% carrageenan, and 0.02% NaN$_3$) followed by overnight incubation with first antibody (1:200) at 4 °C. Then the sections were washed and incubated in biotinylated goat anti-rabbit IgG (1:200; Vector Laboratories, USA), then ABC-Elite (1:200; Vector laboratories, USA) and visualized with diaminobenzidine (1:10,000; Wako, Japan) containing 1% Nickel sodium ammonium and 0.0003% H$_2$O$_2$ in Tris-buffered saline. The adjacent sections were processed with 1% Cresyl Violet.

**Reconstruction of the brain structures.** The line drawings of brain structures in the figures were basically constructed from the MRI images. Ambiguous structures in MRI image (for example, bSC) were reconstructed with reference to the histological sections adjacent to that MRI section (see Supplementary Figure 6).

**Statistics.** All the averaged data in the graphs are stated as mean and SD (Figs. 1d, e, 3d–g, 6c, e, i, k, and 7g). The number of data averaged in Fig. 1d (contralesional (red)) was 105 (21 targets, repeated 5 days). Fifteen for Fig. 1d (ipsilesional (blue)), 80 for Fig. 1e (contralesional (red)), 100 for Fig. 1e (ipsilesional (blue)), and 20 for Fig. 1e (240° (gray)). The statistical evaluations were performed using scilab (Scilab Enterprises, France) with two-tailed unpaired Student's $t$ test for comparisons of data before and after the injection of muscimol (Fig. 3d–g) and number of cells on contralesional and ipsilesional sides (Fig. 7g). In Fig. 6c–l and Supplementary Figure 7, one-tailed unpaired Student's $t$ test for comparison of data before and after the administration of Dox and the interpolation for the $P$ value maps (Fig. 6d, f, j, k and Supplementary Figure 7) were performed by using MATLAB (Mathworks Inc., USA; scatteredInterpolant function was used for the interpolated map).

**Reporting summary.** Further information on experimental design is available in the Nature Research Reporting Summary linked to this article.

## Data availability
The data and codes supporting the findings of this study are available from corresponding author upon reasonable request.

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

## Acknowledgements

This study was supported by Grant-in-Aid for Scientific Research by MEXT to T.I. (Grant Nos. 26112008, 26221003), to H.O. (No. 26112003) and to M.K. (Nos. 25430023, 15H01411, 16K06983). We thank M. Kawahara, M. Togawa, Y. Yamanishi, N. Taka-hashi, K. Shimizu, and K. Takada for technical support and M. Yoshida for conducting the surgery on Monkey-C.

## Author contributions

M.K., R.K., H.O., and T.I. designed the experiments. M.K. and T.I. wrote the paper. M.K. conducted the surgery, behavioral and electrophysiological experiments, and analyzed the data. R.K. and T.I. conducted the surgery. R.K. contributed to the behavioral and electrophysiological experiments and data analysis. K.I. conducted the histological experiments. Kenta Kobayashi and Kazuto Kobayashi made the viral vectors. H.O. contributed to the surgery and behavioral experiment.

## Additional information

**Competing interests:** The authors declare no competing interests.

