## [Peer Review File · Nature Communications]

Reviewers' Comments:

Reviewer #1:

Remarks to the Author:

Major claims of the paper

Using behaving macaques as models, the authors made two claims about a controversial subject in systems neuroscience: the possible pathways mediating "blindsight" (the ability for monkeys and humans to perform visual functions following the lesion of the primary thalamocortical pathway):

Claim 1: The pulvinar nucleus (PUL) is involved in blindsight, as demonstrated by the classical inactivation technique.

Claim 2: Neurons that project from the superior colliculus (SC) to the pulvinar pathway are involved in blindsight, as demonstrated by a novel application of a virus vector that can inactivate specific pathways.

Are the claims novel?

They are not conceptually novel. The SC-to-PUL pathway is the "classical" hypothesis about blindsight. But it has been technically very difficult to test this hypothesis. To do so, the experimental strategy used in this study is clever and novel. The virus vector used is a powerful tool for dissecting neural pathways. This is the first time that it is used to study blindsight in behaving monkeys.

Will the paper be of interest to others in the field?

Yes. It's of interest to those who study neural plasticity, psychology and consciousness. The pulvinar is one of the least studied major structures of the visual system. This study also contribute to the understanding of its functions.

Will the paper influence thinking in the field?

Yes. A competing hypothesis involving the LGN (Schmid et al., 2010) has received significant attention in recent years. An important question that urgently needs to be answered is if the solution is one of the two. Although this study does not necessarily disprove the LGN theory, it strongly suggests that the LGN theory is not sufficient. This study significantly narrows down the possibilities.

Are the claims appropriately discussed in the context of previous literature?

Although the manuscript does establish the context of the large and complex literature, I feel that the introduction and discussion can be edited to be easier to read, more comprehensive, and more insightful. This is a minor criticism.

Are the claims convincing?

Claim 1: Additional data are needed (see below) to demonstrate that the muscimol injections were indeed in the pulvinar. But in general, this claim is convincing.

Claim 2 is not entirely convincing. The double labelling strategy did work to some extent, but not well enough. Figure 6 and S7 show that following the administration of Dox, the error rates of eye saccades were significantly reduced, but only in some of the targets (28% in Monkey C and 50% in

Monkey H, using the criteria chosen by the authors). There doesn't seem to be a pattern among the affected sites. Two adjacent targets produced different results. In addition, in Monkey C, the same procedure repeated twice 3 months apart produced different patterns. This is unexpected given that viruses were injected into large proportions of two retinotopic structures. This might be due to the non-uniform distribution of doubly infected neurons in SC, but there are other possibilities (injection leaked to the white matter, for example). Unless the authors can provide some data (perhaps histological) to explain the patterns in 6C,H and in S7, I am not fully convinced that the results demonstrated that disabling the SC-to-PUL connections diminishes blindsight.

Are there other experiments that would strengthen the paper further? How much would they improve it, and how difficult are they likely to be?

Since both monkeys were killed, suggesting new experiments would be counterproductive. This type of research is very difficult and time consuming.

My concerns raised above might be addressed by more detailed data analysis. The analyses presented in Figure 6 are rudimentary. The metric used ("n" in the insets) is odd. The analyses also did not examine how stimulus contrast affected performance (except for the analysis performed for a single site shown in Fig S6). Perhaps patterns can emerge with deeper data analysis.

If the manuscript is unacceptable in its present form, does the study seem sufficiently promising that the authors should be encouraged to consider a resubmission in the future?

Yes. If Claim 2 could be demonstrated more convincingly, this would be an important contribution to the study of blindsight and thalamocortical pathways. The double labelling methodology is a very powerful tool if it can work effectively.

Detailed comments:

1. The introduction outlines the major hypotheses for blindsight pathways. I think the organization is a bit too messy and can be made more clear.
2. A major issue about this manuscript: the key issue is if the injections were successfully made in the intended structures. This manuscript provides too few anatomical data to illustrate the spatial relationships between the injection sites and the thalamic nuclei. The authors should provide histological sections to complement the MRI scans. Multiple, consecutive sections showing SC, PUL and LGN in both animals should be included. Injection tracks (not just sites) should be clearly visible. Drawings should be accompanied by histological or MRI sections so that the boundaries of the structures can be independently verified by the readers.
3. Figure 2: How were the structures determined? The boundary of bSC is completely invisible in the image provided. How do I know there really is a boundary? What is the red arrow?
4. Figure 3: How many trials of eye saccades were made for each target? The caption does not explain all the features of the figure. What do the dashed circles in A and B mean? What about the crosses and small circles?
5. It's never explained how reaction time is measured.
6. Figure S5: In A, the blue dots are not consistent with the boundary of the MGN. The green diamonds represent sites in the LGN, but I don't see any green diamonds in A and C.

7. In Discussions, the interpretation of the results in the context of Schmid et al (2010) is why this paper is important, and indeed the authors devoted an entire paragraph to it. However, the argument rests on the idea that the LGN of the monkeys in Schmid et al (2010) was not degenerated because the lesion did not remove the white matter. This is a crucial point. It should not be left as a speculation. The authors should confirm this point with the NIH team that published the 2010 paper.

Reviewer #2:

Remarks to the Author:

This manuscript by Kinoshita and colleagues aims to address the role of the ventral Pulvinar in blindsight following V1 lesions. This is an important question that has been investigated in multiple laboratories in both primate and human. In this study 2 monkeys receive lesions to V1 to create hemianopia but with some ability to detect visual targets. Then either the pulvinar or SC is inactivated and the authors demonstrate that this interferes with saccades to targets in the blind field.

While the techniques that have been used are impressive, there is a major flaw in the interpretation. The major issue is that the authors state that visually-guided saccades are impaired by inactivation of the ventral Pulvinar and superior colliculus. However, both of these areas have been implicated in generation of saccades (e.g. Robinson et al (1986) Saccade-related and visual activities in the pulvinar nuclei of the behaving rhesus monkey. *Exp Brain Res* 62:625-34) and the SC in target selection (Horwitz & Newsome (2001) Target selection for saccadic eye movements: prelude activity in the superior colliculus during a direction-discrimination task. *J. Neurophysiol.* 86:2543-58). Thus, it's not clear to me how the experiments can relate to visual function if the response (saccadic eye movements) are impacted upon the inactivation.

To be a 'real' test for blindsight and consider the critical structures for vision it is important to pull apart the response. This could be achieved by using a different type of behavioural response that was not controlled by the SC and pulvinar.

Major issues

1. The authors have ignored much the literature on human blindsight and in marmosets that has been undertaken to try to determine the pathways underlying residual vision. Instead, the one-third of the references are citations of the authors' own work. While introductions and discussions cannot cite all relevant papers, the current manuscript has ignored seminar work from Bourne (marmoset) and Bridge (human). The literature on the roles of SC and pulvinar in visual saccades mentioned above has also been missed and is critical to the interpretation.

2. On lines 72/73 the authors suggest the alternative pathway is SC->dLGN->MT. This is incorrect – the hypothesis is that ganglion cells project to dLGN and then MT.

3. The data here do not appear to provide stronger evidence than the seminal work of Schmid et al cited in the manuscript supporting a critical role for LGN in residual vision.

Minor comments

1. Show the actual MRI scans rather than drawings to allow real inspection of the slices.
2. The paper needs to be rewritten as it is very difficult to follow which makes the work becomes difficult to assess.

Reviewer #3:

Remarks to the Author:

The paper by Kinoshita and colleagues presents a very elegant experiment on the role of the pulvinar and superior colliculus in mediating visually-guided saccades (VGS) in monkeys with V1 damage and "blindsight" (i.e. spared ability to respond to stimuli presented in the affected portion of the visual field).

Monkeys' proficiency to perform saccades in the blind field was impaired by pharmacological inactivation on the visually responsive part of the pulvinar (the ventro-lateral Pulv). Moreover, selective blockade of the connections between the superficial layers of the superior colliculus and the same vIPulv further dropped down precision and latency of saccades toward targets in the contralesional field.

This is a direct and clearly-cut evidence on the critical role of SC-Pulv to some manifestations of blindsight in monkeys, such as VGS. The study is timely and interesting, as it clarifies underlinings of visually guided behaviors in the absence of V1 and, more broadly, of non-conscious visual functions. It is also directly relevant for the understanding of similar phenomena in humans with blindsight, therefore integrating cross-species comparisons, functional neuroanatomy and behavior.

The topic is highly significant and still controversial, thus making new and solid empirical evidence highly welcomed. The paper is generally well-written, methodologically sound, and conclusions are justified. It seems easy to predict that the paper, if published, will likely garner a lot of attention and kindle further thoughts and empirical inquiries on what is already a thriving theme.

I have several comments to offer, which are mainly aimed at broadening the thinking on blindsight on more contemporary bases and theoretical ground, and some minor points I would like the authors to address.

First, the ongoing debate on the neural bases of blindsight in humans and monkeys is partly ill-posed. In fact, it is becoming clear that blindsight is better conceived as a constellation of abilities persisting after V1 damage, rather than as a unitary phenomenon. It follows that multiple pathways can sustain different blindsight functions, which should be better qualified based on specific task demands and stimulus attributes. Authors can find this argument spelled out in more details in Tamietto & Morrone, *Curr Biol* 2016, for example, and an influential taxonomy of different blindsight functions by task and stimuli can be found in Dankert & Rossetti, 2005 *Neurosci & BioBehav Rev*. I am impressed by the present results and I am broadly sympathetic with the view that assigns a pivotal role to the SC and Pulv in blindsight. However, this should not be necessarily considered to contradict prior evidence focusing on the LGN, where the same logical mistake somehow lingering in the present ms is often present; i.e. considering blindsight as unitary and with a unique neural substrate independently of tasks and stimulus type. For example, the fMRI evidence by Schmid et al. 2010 *Nature* was gathered using rotating checkerboards (i.e. moving stimuli). In humans, the most recent and convincing evidence about the primacy of LGN-MT connections in blindsight is also based on the use of moving stimuli (Ajina et al. 2015 *eLife* for anatomical evidence; and Ajina et al., 2018 *PLOS Biol* for functional fMRI). A recent quantitative meta-analysis that considered all published fMRI data on blindsight found indeed that SC, Pulv, LGN, MT and other extra striate areas such as MT, or LIP significantly tilt out, but also that the likelihood to find them active depends of the stimuli used and the tasks employed (Celeghein et al., 2018 *Neuropsychologia*). As the use of moving stimuli and direct guessing in other studies may have resulted in LGN-MT taking the leading role, the use of VGS may unveil a critical role for the SC-Pulv pathway in this task and for basic stimuli. This also offers an additional key to

interpret apparent discrepancies with Schmid et al Nature paper that the authors comment in the discussion.

The sentence in the intro "the contribution of pul to blindsight has never been directly tested" is perhaps a bit ungenerous toward previous research, albeit my comments are not meant to detract value and novelty to the present study. For example, there are very solid evidence on the role of pulvina-MT connections in the V1-damaged marmoset (Warner et al., 2015 Curr Biol, further reviewed in Bridge et al., 2015 TiCS). In human blindsight, there is tractography evidence that the vIPulv is a relay station of a pathway that conveys visual information about saliency from the SC to the amygdala bypassing V1 (Tamietto et al., 2012 Curr Biol), an evidence replicated in macaques (Rafal et al, 2015 J Neurophysiol). On a closely related ground, Pulv neurons respond very early, earlier than V1, to salient stimuli like faces or snakes in macaques (e.g., Van Le et al., 2013 PNAS, Maior et al., 2010 Behav Brain Res).

The authors focus, also on the title, on the pulv, which is fully understandable. However, I see as equally important the second part of their results, namely that blocking the SC-Pulv pathway further worsen VGS. This provides a solid foundation for the fact that the Pulv role in VGS is primarily driven by SC. I would therefore suggest to give equal visibility to this aspect and to the SC, also in the title (e.g., "Dissecting the circuit for blindsight: Critical role of superior colliculus and pulvina" or alike). Still on this ground, authors can find direct evidence in human blindsight that "blocking" the SC abolishes other visuomotor tasks (e.g., Tamietto et al., 2010 JoCN), or affective processing (e.g, Burra et al., 2017 Neuropsychologia), or that blindsight functions are found in patients with hemispherectomy, in whom the only remaining cortical and subcortical structure on one side is the SC (e.g., Leh et al., 2010 JoCN; Georgy et al., 2016 Cortex). This last evidence on hemispherectomy also suggest that the SC is capable of more sophisticated visual function than previously thought, such as face perception, figure-ground segmentation or perceptual grouping.

Finally, in the Discussion the authors mention plasticity incurring after damage that includes white matter underneath V1, as in patient GY. It is probably worth mentioning that this plasticity may lead to the strengthening of existing subcortical-cortical pathways bypassing V1 or the formation of entirely new ones, as reported in papers I already cited above, but also to more extensive rebalancing of the visuomotor functions across hemispheres, involving the dorsal stream, premotor areas and corpus callosum, as recently shown in patient GY (Celegnin et al., 2017 PNAS).

METHODS

The impact of the lesion on vision was assessed with a VGS task identical to the one also used to assess blindsight functions and the impact of pulv and sc-pulv lesions on the same task. I understand the methodological difficulties in training monkeys with multiple tasks, but in principle I would have preferred to see the impact of the lesion on the visual field to be assessed with a perimetry task (yes/no detection without saccades). This is not a major drawback, but implies a certain circularity between how the "blind" and the "sight" aspects of blindsight are assessed, and a cautious remark should be made (better examples on this respect are cited by the authors; ref 25, 26).

In the assessment of visual field defect in relation to saccades (p. 4 first paragraph and fig 1 D and E) the authors report a reduction of accuracy at lower contrast values. Then, I do not see anywhere which contrast values were used for the VGS after Pulv lesion and SC-Pulv inactivation. Please specify.

It may not be obvious to readers from other backgrounds in neuroscience why CaMKII is good to detect K cells. Good to quote Sinchic et al., 2004 Nat Neurosci as the author did, but probably a more

direct reference to this is Hendry & Yoshioka 1994 Science.

WRITING and TYPOS

The ms is generally well-written and clear. A relative exception is the abstract, whereby several sentences need polishing. In UPPERCASE my suggestions, focusing only on main aspects.

"In patients WITH damage to the primary visual cortex (V1), residual vision can guide goal-directed movements to targets in the blind field without awareness. This phenomenon HAS BEEN termed 'blindsight', AND ITS neural mechanismS ARE controversial. ...[] ...Next, selective and reversible blockade of the SC-vIPul pathway by combining two viral vectors FURTHER impaired VGS COMPARED TO INACTIVATION OF vIPULV ALONE.

In the last line of the Discussion I would avoid the use of "subconscious", which traditionally hinges on psychodynamic and psychoanalytical connotations, in favor of a more standard and neuroscientific common "non-conscious".

Summing up, this is an elegant study and important contribution to the literature on blindsight and on V1-independent visual functions. The paper is already in good shape and, IMHO, it would deserve publication in Nature Communications. Besides minor clarifications, my main request to the authors is to cast their results better in the context of previous studies on blindsight, pulvinar and superior colliculus in humans and monkeys, and in acknowledging more clearly the multifaceted nature of the phenomenon and its neural underpinnings.

According to the Journal suggestion and policy, I sign my review.

Reviewer #1 (Remarks to the Author):

Major claims of the paper

Using behaving macaques as models, the authors made two claims about a controversial subject in systems neuroscience: the possible pathways mediating "blindsight" (the ability for monkeys and humans to perform visual functions following the lesion of the primary thalamocortical pathway):

Claim 1: The pulvinar nucleus (PUL) is involved in blindsight, as demonstrated by the classical inactivation technique.

Claim 2: Neurons that project from the superior colliculus (SC) to the pulvinar pathway are involved in blindsight, as demonstrated by a novel application of a virus vector that can inactivate specific pathways.

Are the claims novel?

They are not conceptually novel. The SC-to-PUL pathway is the "classical" hypothesis about blindsight. But it has been technically very difficult to test this hypothesis. To do so, the experimental strategy used in this study is clever and novel. The virus vector used is a powerful tool for dissecting neural pathways. This is the first time that it is used to study blindsight in behaving monkeys.

Will the paper be of interest to others in the field?

Yes. It's of interest to those who study neural plasticity, psychology and consciousness. The pulvinar is one of the least studied major structures of the visual system. This study also contribute to the understanding of its functions.

Will the paper influence thinking in the field?

Yes. A competing hypothesis involving the LGN (Schmid et al., 2010) has received significant attention in recent years. An important question that urgently needs to be answered is if the solution is one of the two. Although this study does not necessarily disprove the LGN theory, it strongly suggests that the LGN theory is not sufficient. This study significantly narrows down the possibilities.

Are the claims appropriately discussed in the context of previous literature?

Although the manuscript does establish the context of the large and complex literature, I feel that the introduction and discussion can be edited to be easier to read, more comprehensive, and more insightful. This is a minor criticism.

Our response; Thank you for the suggestions. We extensively revised both Introduction and Discussion with additional references.

Are the claims convincing?

Claim 1: Additional data are needed (see below) to demonstrate that the muscimol injections were indeed in the pulvinar. But in general, this claim is convincing.

Claim 2 is not entirely convincing. The double labelling strategy did work to some extent, but not well enough. Figure 6 and S7 show that following the administration of Dox, the error rates of eye saccades were significantly reduced, but only in some of the targets (28% in Monkey C and 50% in Monkey H, using the criteria chosen by the authors). There doesn't seem to be a pattern among the affected sites. Two adjacent targets produced different results. In addition, in Monkey C, the same procedure repeated twice 3 months apart produced different patterns. This is unexpected given that viruses were injected into large proportions of two retinotopic structures. This might be due to the non-uniform distribution of doubly infected neurons in SC, but there are other possibilities (injection leaked to the white matter, for example). Unless the authors can provide some data (perhaps histological) to explain the patterns in 6C,H and in S7, I am not fully convinced that the results

demonstrated that disabling the SC-to-PUL connections diminishes blindsight.

Our response; Thank you for the suggestions. According to your remarks below (about the analysis and metric used in Figure 6), we revised the analysis method and Figure 6. In the revised Figure 6, we showed the map of P value of standard statistical test for two indices (the size of saccade error and the reaction time). As you mentioned, the effect of double infection method did not cover all the tested target locations, however, affected locations were largely overlapped between the two indices and across different target contrasts (new Figure S7). This may indicate the double infection method worked, at least partially.

Before 2nd Dox administration experiment, which was two months after termination of 1st Dox administration, the VGS performance of the animal became considerably deteriorated, not just at the Dox-affected target locations during the 1st administration but also other target areas (the overall success ratio of VGS declined from 60-80% (1st round) to 40% and fluctuating (2nd round), which was too poor for ordinary analysis of saccade performance). The reason of such low performance was likely to be due to bad physical condition of the monkey. Because the data taken from the animal which was suspected to be in bad physical condition should not be used, we have decided to retract Supplementary Figures S6 and S7, which contained the data from the 2nd Dox experiment, from the revised manuscript. Instead, to show the consistency of the results, we added new Figure S7 which shows the effect of target contrasts.

Are there other experiments that would strengthen the paper further? How much would they improve it, and how difficult are they likely to be?

Since both monkeys were killed, suggesting new experiments would be counterproductive. This type of research is very difficult and time consuming.

My concerns raised above might be addressed by more detailed data analysis. The analyses presented in Figure 6 are rudimentary. The metric used ("n" in the insets) is odd. The analyses also

did not examine how stimulus contrast affected performance (except for the analysis performed for a single site shown in Fig S6). Perhaps patterns can emerge with deeper data analysis.

If the manuscript is unacceptable in its present form, does the study seem sufficiently promising that the authors should be encouraged to consider a resubmission in the future?

Yes. If Claim 2 could be demonstrated more convincingly, this would be an important contribution to the study of blindsight and thalamocortical pathways. The double labelling methodology is a very powerful tool if it can work effectively.

Our response; Thank you for the suggestions. In response to your remarks, we revised the analysis method and Figure 6. We revised also the texts in the Results and added new Supplementary Fig. S7.

In the original Figure 6, we used the success ratio as one index. Because of the small number of data (success ratio is a single value per day), a power of statistical test was not so strong. In the revised Figure 6, instead of the success ratio, we analyzed the size of saccade error, which we have multiple values per day (= number of trials) and used standard statistical test (t-test). We also revised the effect maps with interpolated maps for ease to see the affected field. To answer ‘*how stimulus contrast affected performance*’, we added new Figure S7, in which the saccade error was indicated for various target contrast.

Detailed comments:

1. The introduction outlines the major hypotheses for blindsight pathways. I think the organization is a bit too messy and can be made more clear.

Our response; Thank you for the suggestions. We revised Introduction with additional references.

2. A major issue about this manuscript: the key issue is if the injections were successfully made in the intended structures. This manuscript provides too few anatomical data to illustrate the spatial relationships between the injection sites and the thalamic nuclei. The authors should provide histological sections to complement the MRI scans. Multiple, consecutive sections showing SC, PUL and LGN in both animals should be included. Injection tracks (not just sites) should be clearly visible. Drawings should be accompanied by histological or MRI sections so that the boundaries of the structures can be independently verified by the readers.

Our response; Thank you for the suggestions. We added the series of Nissl-stained histological sections (with some tracks of injection needles) and the MRI scans to Fig. S6 showing SC, Pul, LGN for both monkeys. We also added the Nissl-stained section showing the injection track in Fig. 5C.

3. Figure 2: How were the structures determined? The boundary of bSC is completely invisible in

the image provided. How do I know there really is a boundary? What is the red arrow?

Our response; Thank you for the suggestions. We added a section termed “Reconstruction of the brain structures” in Methods. The line drawings were basically made from the MRI image, however, structures unclear in MRI image were reconstructed with reference to the histological sections. We also added the histological section adjacent to the MRI image and description about the red arrowhead in Figure 2.

4. Figure 3: How many trials of eye saccades were made for each target? The caption does not explain all the features of the figure. What do the dashed circles in A and B mean? What about the crosses and small circles?

Our response; Thank you for the suggestions. We added explanations about the number of trials in captions of Figure 3 and S2 and added explanations about the marks in caption of Figure 3A and B.

5. It's never explained how reaction time is measured.

Our response; Thank you for the suggestions. We added brief explanation of the reaction time in the ‘Pulvinar inactivation’ section of Results and detailed explanation in the ‘Task’ section of Methods.

6. Figure S5: In A, the blue dots are not consistent with the boundary of the MGN. The green diamonds represent sites in the LGN, but I don't see any green diamonds in A and C.

Our response; Thank you for the suggestions. We fixed the descriptions about the locations of sound- or visual-responsive neurons in caption and inset of Fig. S6, because detailed tests to identify the recorded cells or axons as belonging to the MGN or LGN were not performed. We were not sure why the sound responsive activities were recorded from outside the MGN. One possible reason might be that we recorded from passing fibers of the MGN, or some Pul neurons could be related to auditory sensation. For the green symbols, we fixed our mistake in the previous version which you pointed out. We added the missed symbols in Figure S6G (A8, Monkey-H). (These symbols were missed during the operation of drawing software to print on the bitmap picture)

7. In Discussions, the interpretation of the results in the context of Schmid et al (2010) is why this paper is important, and indeed the authors devoted an entire paragraph to it. However, the argument rests on the idea that the LGN of the monkeys in Schmid et al (2010) was not degenerated because the lesion did not remove the white matter. This is a crucial point. It should not be left as a speculation. The authors should confirm this point with the NIH team that published the 2010 paper.

Our response; Thank you for the suggestions. we contacted Michael Schmid and he sent us a

figure which was not included in his paper in 2010. The figure showed clear activation of dLGN in response to the moving stimulus in the bold fMRI recordings. Therefore, we referred to his result as personal communication. In addition, Michael sent us a figure of the histology taken more than 3 years after the data for the 2010 paper, showing that some MT-projecting neurons (retrogradely labeled from MT) were still remaining, but this was taken some years later and not quantitative, we decided not to refer to the results.

Reviewer #2 (Remarks to the Author):

This manuscript by Kinoshita and colleagues aims to address the role of the ventral Pulvinar in blindsight following V1 lesions. This is an important questions that has been investigated in multiple laboratories in both primate and human. In this study 2 monkeys receive lesions to V1 to create hemianopia but with some ability to detect visual targets. Then either the pulvinar or SC is inactivated and the authors demonstrate that this interferes with saccades to targets in the blind field.

While the techniques that have been used are impressive, there is a major flaw in the interpretation. The major issue is that the authors state that visually-guided saccades are impaired by inactivation of the ventral Pulvinar and superior colliculus. However, both of these areas have been implicated in generation of saccades (e.g. Robinson et al (1986) Saccade-related and visual activities in the pulvinar nuclei of the behaving rhesus monkey. Exp Brain Res 62:625-34) and the SC in target selection (Horwitz & Newsome (2001) Target selection for saccadic eye movements: prelude activity in the superior colliculus during a direction-discrimination task. J. Neurophysiol. 86:2543-58). Thus, it's not clear to me how the experiments can relate to visual function if the response (saccadic eye movements) are impacted upon the inactivation.

Our response;

The question in this paper was the neural systems for the control of visually guided saccades in blindsight. In original definition by Weiskrantz, blindsight is the dissociation between the verbal report of “seeing” and forced choice “guessing”, that is, “non-conscious visuo-motor processing observed after lesion of the visual cortex”. We found that the superior colliculus (SC) and ventral pulvinar (Pul) are critical for the control of visually guided saccades in blindsight animals. As detailed in the revised version of Introduction and Discussion, blindsight would not be a unitary phenomenon, but would be a constellation of functions persisting after the lesion of V1, that is, multiple pathways can sustain different blindsight functions. From the literature, it appears that when responses to moving visual stimulus was measured, LGN-MT pathway tended to be activated, while in case of visually guided saccades to stationary targets, the SC-Pul pathway would play more significant roles, as suggested in this paper. Thus, visual ability would be an element of blindsight, but visuo-motor processing for saccade generation is surely another critical element of blindsight. Therefore, we think there is no flaw of interpretation.

Many previous studies showed that inactivation of SC in intact monkeys delays latency of saccades but does not impair execution of saccades. (We have preliminary data showing that inactivation of vIPul in intact monkeys do not impair saccade execution, either.) Therefore, the present results (inactivation of tecto-recipient zone of the Pul, which has been shown to carry saccade-related visual signals in previous studies) cannot be interpreted as the results of impairment of saccade motor system.

To be a 'real' test for blindsight and consider the critical structures for vision it is important to pull apart the response. This could be achieved by using a different type of behavioural response that was not controlled by the SC and pulvinar.

Major issues

- 1. The authors have ignored much the literature on human blindsight and in marmosets that has been undertaken to try to determine the pathways underlying residual vision. Instead, the one-third of the references are citations of the authors' own work. While introductions and discussions cannot cite all relevant papers, the current manuscript has ignored seminal work from Bourne (marmoset) and Bridge (human). The literature on the roles of SC and pulvinar in visual saccades mentioned above has also been missed and is critical to the interpretation.*

Our response;

We appreciate the comment. By taking into account of the comments also from other two reviewers, we extensively revised Introduction and Discussion with relevant references.

- 2. On lines 72/73 the authors suggest the alternative pathway is SC->dLGN->MT. This is incorrect – the hypothesis is that ganglion cells project to dLGN and then MT.*

Our response;

Actually, Schmid et al. (2010) proposed that the alternative pathway is SC-dLGN-MT. But all the relevant articles referred to in this text are not making this proposition, we limited ourselves to mention about the role of LGN-MT pathway in blindsight.

- 3. The data here do not appear to provide stronger evidence than the seminal work of Schmid et al cited in the manuscript supporting a critical role for LGN in residual vision.*

Our response;

We are not excluding the LGN hypothesis. For that, we need a double dissociation experiments in single animals as we argued in the latter part of Discussion. But to propose the role of Pul and SC-Pul pathway in blindsight, we believe the content of this article is strong enough.

Minor comments

- 1. Show the actual MRI scans rather than drawings to allow real inspection of the slices.*

Our response;

Thank you for the suggestion. We added not only the actual MRI scans but also the Nissl-stained histological sections in Figs. 5 and S6 as suggested by Reviewer #1. We also added Supplementary Fig. S1 to complement the drawings in Fig. 1A.

2. *The paper needs to be rewritten as it is very difficult to follow which makes the work becomes difficult to assess.*

Our response;

Sorry for the poor readability of the original text. We extensively revised the text.

Reviewer #3 (Remarks to the Author):

The paper by Kinoshita and colleagues presents a very elegant experiment on the role of the pulvinar and superior colliculus in mediating visually-guided saccades (VGS) in monkeys with V1 damage and "blindsight" (i.e. spared ability to respond to stimuli presented in the affected portion of the visual field).

Monkeys' proficiency to perform saccades in the blind field was impaired by pharmacological inactivation on the visually responsive part of the pulvinar (the ventro-lateral Pulv). Moreover, selective blockade of the connections between the superficial layers of the superior colliculus and the same vlPulv further dropped down precision and latency of saccades toward targets in the contralesional field.

This is a direct and clearly-cut evidence on the critical role of SC-Pulv to some manifestations of blindsight in monkeys, such as VGS. The study is timely and interesting, as it clarifies underlinings of visually guided behaviors in the absence of V1 and, more broadly, of non-conscious visual functions. It is also directly relevant for the understanding of similar phenomena in humans with blindsight, therefore integrating cross-species comparisons, functional neuroanatomy and behavior. The topic is highly significant and still controversial, thus making new and solid empirical evidence highly welcomed. The paper is generally well-written, methodologically sound, and conclusions are justified. It seems easy to predict that the paper, if published, will likely garner a lot of attention and kindle further thoughts and empirical inquires on what is already a thriving theme.

I have several comments to offer, which are mainly aimed at broadening the thinking on blindsight on more contemporary bases and theoretical ground, and some minor points I would like the authors to address.

First, the ongoing debate on the neural bases of blindsight in humans and monkeys is partly ill-posed. In fact, it is becoming clear that blindsight is better conceived as a constellation of abilities persisting after V1 damage, rather than as a unitary phenomenon. It follows that multiple pathways can sustain different blindsight functions, which should be better qualified based on specific task demands and stimulus attributes. Authors can find this argument spelled out in more details in Tamietto & Morrone, Curr Biol 2016, for example, and an influential taxonomy of different blindsight functions by task and stimuli can be found in Dankert & Rossetti, 2005 Neurosci & BioBehav Rev. I am impressed by the present results and I am broadly sympathetic with

the a view that assigns a pivotal role to the SC and Pulv in blindsight. However, this should not be necessarily considered to contradict prior evidence focusing on the LGN, where the same logical mistake somehow lingering in the present ms is often present; i.e. considering blindsight as unitary and with a unique neural substrate independently of tasks and stimulus type. For example, the fMRI evidence by Schmid et al. 2010 Nature was gathered using rotating checkerboards (i.e. moving stimuli). In humans, the most recent and convincing evidence about the primacy of LGN-MT connections in blindsight is also based on the use of moving stimuli (Ajina et al. 2015 eLife for anatomical evidence; and Ajina et al., 2018 PLOS Biol for functional fMRI). A recent quantitative meta-analysis that considered all published fMRI data on blindsight found indeed that SC, Pulv, LGN, MT and other extra striate areas such as MT, or LIP significantly tilt out, but also that the likelihood to find them active depends of the stimuli used and the tasks employed (Celeghein et al., 2018 Neuropsychologia). As the use of moving stimuli and direct guessing in other studies may have resulted in LGN-MT taking the leading role, the use of VGS may unveil a critical role for the SC-Pulv pathway in this task and for basic stimuli. This also offers an additional key to interpret apparent discrepancies with Schmid et al Nature paper that the authors comment in the discussion.

Our response; Thank you for the suggestions. I fully agree with the concept that blindsight should be considered as a constellation of abilities persisting after V1 damage, rather than as a unitary phenomenon, but multiple pathways can sustain different blindsight functions. I revised both Introduction and Discussion in this context with reference to the suggested articles.

The sentence in the intro "the contribution of pul to blindsight has never been directly tested" is perhaps a bit ungenerous toward previous research, albeit my comments are not meant to detract value and novelty to the present study. For example, there are very solid evidence on the role of pulvinar-MT connections in the V1-damaged marmoset (Warner et al., 2015 Curr Biol, further reviewed in Bridge et al., 2015 TiCS). In human blindsight, there is tractography evidence that the vlPulv is a relay station of a pathway that conveys visual information about saliency from the SC to the amygdala bypassing V1 (Tamietto et al., 2012 Curr Biol), an evidence replicated in macaques (Rafal et al, 2015 J Neurophysiol). On a closely related ground, Pulv neurons respond very early, earlier than V1, to salient stimuli like faces or snakes in macaques (e.g., Van Le et al., 2013 PNAS, Maior et al., 2010 Behav Brain Res).

Our response; Thank you for the suggestions. We still think that the contribution of Pul to blindsight has never been directly tested in our context. But we fully agree with your suggestion that not referring to these relevant articles is not fair. We referred to these relevant articles suggesting the contribution of Pul to blindsight.

The authors focus, also on the title, on the pulv, which is fully understandable. However, I see as equally important the second part of their results, namely that blocking the SC-Pulv pathway further worsen VGS. This provides a solid foundation for the fact that the Pulv role in VGS is primarily driven by SC. I would therefore suggest to give equal visibility to this aspect and to the SC, also in the title (e.g., "Dissecting the circuit for blindsight: Critical role of superior colliculus and pulvinar" or alike).

Our response; Thank you for the suggestion. We changed the title as suggested.

Still on this ground, authors can find direct evidence in human blindsight that "blocking" the SC abolishes other visuomotor tasks (e.g., Tamietto et al., 2010 JoCN), or affective processing (e.g., Burra et al., 2017 Neuropsychologia), or that blindsight functions are found in patients with hemispherectomy, in whom the only remaining cortical and subcortical structure on one side is the SC (e.g., Leh et al., 2010 JoCN; Georgy et al., 2016 Cortex). This last evidence on hemispherectomy also suggest that the SC is capable of more sophisticated visual function than previously thought, such as face perception, figure-ground segmentation or perceptual grouping.

Our response; Thank you for the suggestions. We referred to these articles in Introduction, where we referred to critical involvement of the SC in blindsight.

Finally, in the Discussion the authors mention plasticity incurring after damage that includes white matter underneath VI, as in patient GY. It is probably worth mentioning that this plasticity may lead to the strengthening of existing subcortical-cortical pathways bypassing VI or the formation of entirely new ones, as reported in papers I already cited above, but also to more extensive rebalancing of the visuomotor functions across hemispheres, involving the dorsal stream, premotor areas and corpus callous, as recently shown in patient GY (Celegin et al., 2017 PNAS).

Our response; Thank you for the suggestion. We referred to this article with description about the large scaled network plasticity in blindsight patients.

METHODS

The impact of the lesion on vision was assessed with a VGS task identical to the one also used to assess blindsight functions and the impact of pulv and sc-pulv lesions on the same task. I understand the methodological difficulties in training monkeys with multiple tasks, but in principle I would have preferred to see the impact of the lesion on the visual field to be assessed with a perimetry task (yes/no detection without saccades). This is not a major drawback, but implies a certain circularity between how the "blind" and the "sight" aspects of blindsight are assessed, and a cautious remark should be made (better examples on this respect are cited by the authors; ref 25, 26).

Our response; Thank you for the suggestion. Actually it has been a problem in our series of blindsight studies that we were using the saccade task. In our new ongoing experiments, we added the manual response task. We described the necessity of adopting the manual response task in the place where we mentioned about the future experiments.

In the assessment of visual field defect in relation to saccades (p. 4 first paragraph and fig 1 D and E) the authors report a reduction of accuracy at lower contrast values. Then, I do not see anywhere which contrast values were used for the VGS after Pulv lesion and SC-Pulv inactivation. Please specify.

Our response; Thank you for the suggestion. We added descriptions about target contrast in the figure captions (Fig. 3, 6 and Supplementary Fig. 2).

It may not be obvious to readers from other backgrounds in neuroscience why CaMKII is good to detect K cells. Good to quote Sinchic et al., 2004 Nat Neurosci as the author did, but probably a more

direct reference to this is Hendry & Yoshioka 1994 Science.

Our response; Thank you for the suggestion. We referred to Hendry and Yoshioka (1994).

WRITING and TYPOS

The ms is generally well-written and clear. A relative exception is the abstract, whereby several sentences need polishing. In UPPERCASE my suggestions, focusing only on main aspects.

"In patients WITH damage to the primary visual cortex (V1), residual vision can guide goal-directed movements to targets in the blind field without awareness. This phenomenon HAS BEEN termed 'blindsight', AND ITS neural mechanismS ARE controversial. ...[] ...Next, selective and reversible blockade of the SC-vlPul pathway by combining two viral vectors FURTHER impaired VGS COMPARED TO INACTIVATION OF vlPULV ALONE.

Our response; Thank you for your suggestion. We corrected the abstract according to the suggestion. However, there is one misunderstanding. The pathway-selective blocking experiments were not conducted in addition to the inactivation of vlPul alone. They were conducted as separate experiments. We deleted "further".

In the last line of the Discussion I would avoid the use of "subconscious", which traditionally hinges on psychodynamic and psychoanalytical connotations, in favor of a more standard and neuroscientific common "non-conscious".

Our response; Thank you for the suggestion. We revised as suggested.

Summing up, this is an elegant study and important contribution to the literature on blindsight and on V1-independent visual functions. The paper is already in good shape and, IMHO, it would deserve publication in Nature Communications. Besides minor clarifications, my main request to the authors is to cast their results better in the context of previous studies on blindsight, pulvinar and superior colliculus in humans and monkeys, and in acknowledging more clearly the multifaceted nature of the phenomenon and its neural underpinnings.

Our response; Thank you for all the suggestions and positive comments. We believe the revisions were properly made and the revised manuscript could be acceptable for publication in Nature Communications.

Reviewers' Comments:

Reviewer #1:

Remarks to the Author:

In my review for the previous submission of this work, I raised two major issues: 1. I pointed out that the anatomical locations of the injection sites were not carefully documented, and 2. I believed that the effectiveness of the double-labeling experiment was not convincing. Both issues (and other minor issues) have been addressed by this revision. I believe that the new figures made a convincing case for the role of the SC to PUL pathway in blindsight. However, I still feel that some additional editing is needed, to explain the significance of this work in a more thoughtful way. I would like to ask the authors to address the following issues:

1. The authors had made significant effort to ensure that the injections were limited to the pulvinar. However, the imaging data in Figure 4 still does not completely rule out the possibility that muscimol might have affected the LGN. The imaging method might not be sensitive enough to show muscimol in the LGN. I would like the author to acknowledge in Discussions this possibility.
2. In the reply, the authors stated that data from the second behavior test (in dox administration) of Monkey C was retracted, due to health problems developed after the first behavior test. This can be granted, if 1: the main text of the paper makes a brief statement about why data from the second test is not shown, and 2. A supplementary figure is added to support the claim that the second test from Monkey C can be excluded.
3. I suggested that the authors should contact Schmid about the state of the LGN in their fMRI study. The reply from Schmid I think is inconclusive. Schmid stated that BOLD signal was detected in the LGN of their V1-lesioned macaque. However, this does not mean that the LGN was not degenerated. A degenerated LGN with a small number of survived neurons might still give detectable BOLD signal. I think this comment from Schmid is a non sequitur. It does not make the case for the authors' hypothesis, and therefore should be edited out.
4. The logic of Discussions is muddled, in my opinion. Line 250+ says that blindsight is supported by multiple pathways and task dependent (this I agree). But if that is true, there is no need to explain the result of Schmid in so many words in line 218, because the results are not contradictory. The Discussion I think can be made more concise and more focused. Reviewer 2 also seems to be confused by the author's position with respect to Schmid et al.
5. Reviewer 2 asked if eye saccade was impaired by damages made in the injections in SC. The question is important enough that I think it should be addressed in Discussions. The authors reply to Reviewer 2 I think is not wrong but not in sufficient detail.

Reviewer #2:

Remarks to the Author:

The authors have significantly improved the manuscript, and it reflects the current state of the literature much more accurately now. I do still have a few interpretation comments on the manuscript, however to be added to the Discussion.

1. While the authors have commented on the issue about saccades in the response to reviewers, this has not been adequately addressed in the Discussion. My interpretation of their response is that the authors are testing the visuo-motor response to a visual stimulus, not the unconscious perception of

it. What they have then demonstrated is that the response is disrupted, but they cannot pull apart whether this is related to disruption of the saccade or the 'blindsight' per se.

This needs to be stated explicitly. The authors can use the literature to provide support away from it being saccades, but as far as I can tell, they haven't shown that here. I realise from the author response that this is currently being addressed with their new task, but is not here.

The Discussion needs to be very explicit about this, and ideally if it could be broken down into subsections, it would be easier to extract the major points.

2. Is there a possibility of damage to a pathway between LGN and MT from the V1 lesion? If so, this would obviously impact upon the results and should be included in the Discussion.

Reviewer #3:

Remarks to the Author:

The authors made a thorough job in replying to previous comments.

Improvements are always possible...but I think that this version is solid enough to be put on records and find space in Nat Communications.

Perhaps, a cautionary remark should be made more explicit when referring to the limitations of the present study in the Discussion. It essentially concerns the risk of double-dipping or circularity in two possible aspects. The same task was used to assess the impact of the lesion on (conscious) visual functions as well as the remaining "blindsight" functions. Second, the lesioned structures partly contribute to the execution of the task that was used to evaluate blindsight functions.

I appreciate the authors reply when they say that manual responses should have been a better choice and also when saying that damage to SC or inactivation of dIPulv delays latency of saccades but does not impair its execution. I only think that both points are acknowledged more directly and transparently in the response letter than in the manuscript text, and perhaps should be listed more clearly in the Discussion as current limitations and suggestions for further studies.

NCOMMS-18-20314A

REVIEWERS' COMMENTS:

Reviewer #1 (Remarks to the Author):

In my review for the previous submission of this work, I raised two major issues: 1. I pointed out that the anatomical locations of the injection sites were not carefully documented, and 2. I believed that the effectiveness of the double-labeling experiment was not convincing. Both issues (and other minor issues) have been addressed by this revision. I believe that the new figures made a convincing case for the role of the SC to PUL pathway in blindsight. However, I still feel that some additional editing is needed, to explain the significance of this work in a more thoughtful way. I would like to ask the authors to address the following issues:

1. The authors had made significant effort to ensure that the injections were limited to the pulvinar. However, the imaging data in Figure 4 still does not completely rule out the possibility that muscimol might have affected the LGN. The imaging method might not be sensitive enough to show muscimol in the LGN. I would like the author to acknowledge in Discussions this possibility.

Our response;

We added the statements about the limitation of the visualization method using MRI contrast agent (Discussion 1st paragraph, line 6-11).

2. In the reply, the authors stated that data from the second behavior test (in dox administration) of Monkey C was retracted, due to health problems developed after the first behavior test. This can be granted, if 1: the main text of the paper makes a brief statement about why data from the second test is not shown, and 2. A supplementary figure is added to support the claim that the second test from Monkey C can be excluded.

Our response;

We added (1) statements about the reason of the retraction in ‘Method – Doxycycline administration’ section, and (2) the Supplementary Figure 8 to support the statements.

3. I suggested that the authors should contact Schmid about the state of the LGN in their fMRI study. The reply from Schmid I think is inconclusive. Schmid stated that BOLD signal was detected in the LGN of their V1-lesioned macaque. However, this does not mean that the LGN was not degenerated. A degenerated LGN with a small number of survived neurons might still give detectable BOLD signal. I think this comment from Schmid is a non sequitur. It does not make the case for the authors' hypothesis, and therefore should be edited out.

Our response;

We removed the statement about the personal communication from Schmid from the Discussion (3rd paragraph, line 6).

4. The logic of Discussions is muddled, in my opinion. Line 250+ says that blindsight is supported by multiple pathways and task dependent (this I agree). But if that is true, there is no need to explain the result of Schmid in so many words in line 218, because the results are not contradictory. The Discussion I think can be made more concise and more focused. Reviewer 2 also seems to be confused by the author's position with respect to Schmid et al.

Our response;

We removed the explanation of the result of Schmid et al. at line 218 of 1st revision text (Discussion 3rd paragraph, line 6).

5. Reviewer 2 asked if eye saccade was impaired by damages made in the injections in SC. The question is important enough that I think it should be addressed in Discussions. The authors reply to Reviewer 2 I think is not wrong but not in sufficient detail.

Our response;

We added a new paragraph to discuss on this issue (Discussion, 2nd paragraph).

Reviewer #2 (Remarks to the Author):

The authors have significantly improved the manuscript, and it reflects the current state of the literature much more accurately now. I do still have a few interpretation comments on the manuscript, however to be added to the Discussion.

1. While the authors have commented on the issue about saccades in the response to reviewers, this has not been adequately addressed in the Discussion. My interpretation of their response is that the authors are testing the visuo-motor response to a visual stimulus, not the unconscious perception of it. What they have then demonstrated is that the response is disrupted, but they cannot pull apart whether this is related to disruption of the saccade or the 'blindsight' per se.

This needs to be stated explicitly. The authors can use the literature to provides support away from it being saccades, but as far as I can tell, they haven't shown that here. I realise from the author response that this is currently being addressed with their new task, but is not here.

The Discussion needs to be very explicit about this, and ideally if it could be broken down into subsections, it would be easier to extract the major points.

Our response;

We added a new paragraph to discuss on this issue (Discussion, 2nd paragraph).

2. Is there a possibility of damage to a pathway between LGN and MT from the V1 lesion? If so, this would obviously impact upon the results and should be included in the Discussion.

Our response;

We added the statement about this issue (Discussion 3rd paragraph, line 4-3 from the bottom).

Reviewer #3 (Remarks to the Author):

The authors made a thorough job in replying to previous comments.

Improvements are always possible...but I think that this version is solid enough to be put on records and find space in Nat Communications.

Perhaps, a cautionary remark should be made more explicit when referring to the limitations of the present study in the Discussion. It essentially concerns the risk of double-dipping or circularity in two possible aspects. The same task was used to assess the impact of the lesion on (conscious) visual functions as well as the remaining "blindsight" functions. Second, the lesioned structures partly contribute to the execution of the task that was used to evaluate blindsight functions.

Our response;

Regarding the first point, we modified the statement on this issue (Discussion 4th paragraph last two sentences).

Regarding the second point, if “the lesioned structures” means that the inactivation of SC/Pul, we added a new paragraph to discuss on this issue (Discussion, 2nd paragraph), or if “the lesioned structures” means extensive V1 lesion, we added the statement about the issue of potential damage to the LGN-MT pathway after the V1 lesion (Discussion 3rd paragraph, line 4-3 from the bottom).

I appreciate the authors reply when they say that manual reposes should have been a better choice and also when saying that damage to SC or inactivation of dlPulv delays latency of saccades but does not impair its execution. I only think that both points are acknowledged more directly and transparently in the response letter than in the manuscript text, and perhaps should be listed more clearly in the Discussion as current limitations and suggestions for further studies.

Our response;

We added a new paragraph to discuss on this issue of damage to SC or inactivation of Pul (Discussion, 2nd paragraph) and statement about the current limitations and suggestions (Discussion 3rd paragraph, line 4-3 from the bottom).